# Innovative Research Offers New Hope for Managing African Swine Fever Better in Resource-Limited Smallholder Farming Settings: A Timely Update

**DOI:** 10.3390/pathogens12020355

**Published:** 2023-02-20

**Authors:** Mary-Louise Penrith, Juanita van Heerden, Dirk U. Pfeiffer, Edvīns Oļševskis, Klaus Depner, Erika Chenais

**Affiliations:** 1Department of Veterinary Tropical Diseases, Faculty of Veterinary Science, University of Pretoria, Onderstepoort, Pretoria 0110, South Africa; 2Transboundary Animal Diseases, Onderstepoort Veterinary Research, Agricultural Research Council, Pretoria 0110, South Africa; 3Centre for Applied One Health Research and Policy Advice, Jockey Club College of Veterinary Medicine and Life Sciences, City University of Hong Kong, Kowloon, Hong Kong SAR, China; 4Department of Pathobiology and Population Sciences, Veterinary Epidemiology, Economics, and Public Health Group, Royal Veterinary College, Hatfield AL9 7TA, UK; 5Food and Veterinary Service, LV-1050 Riga, Latvia; 6Institute of Food Safety, Animal Health and Environment, “BIOR“, LV-1076 Riga, Latvia; 7Friedrich-Loeffler-Institute, Greifswald-Insel Riems, 17493 Greifswald, Germany; 8Department of Disease Control and Epidemiology, National Veterinary Institute, S-751 89 Uppsala, Sweden

**Keywords:** African swine fever, smallholder, backyard, pig, biosecurity, control, modified culling

## Abstract

African swine fever (ASF) in domestic pigs has, since its discovery in Africa more than a century ago, been associated with subsistence pig keeping with low levels of biosecurity. Likewise, smallholder and backyard pig farming in resource-limited settings have been notably affected during the ongoing epidemic in Eastern Europe, Asia, the Pacific, and Caribbean regions. Many challenges to managing ASF in such settings have been identified in the ongoing as well as previous epidemics. Consistent implementation of biosecurity at all nodes in the value chain remains most important for controlling and preventing ASF. Recent research from Asia, Africa, and Europe has provided science-based information that can be of value in overcoming some of the hurdles faced for implementing biosecurity in resource-limited contexts. In this narrative review we examine a selection of these studies elucidating innovative solutions such as shorter boiling times for inactivating ASF virus in swill, participatory planning of interventions for risk mitigation for ASF, better understanding of smallholder pig-keeper perceptions and constraints, modified culling, and safe alternatives for disposal of carcasses of pigs that have died of ASF. The aim of the review is to increase acceptance and implementation of science-based approaches that increase the feasibility of managing, and the possibility to prevent, ASF in resource-limited settings. This could contribute to protecting hundreds of thousands of livelihoods that depend upon pigs and enable small-scale pig production to reach its full potential for poverty alleviation and food security.

## 1. Introduction

The global spread of African swine fever (ASF), affecting a wide variety of countries and pig production systems, has confirmed the need to develop prevention and control measures that take into account both the characteristics of the virus and the context in which it affects the host [1]. Smallholder pig holdings with low biosecurity have been over-represented in the reports of ASF from low- and middle-income countries (LMICs), where they often predominate and play an important part in rural agriculture [2,3,4,5,6,7]. In many LMICs, pigs are kept by predominantly economically vulnerable groups, often in traditional, wholly or partially free-ranging husbandry systems. Apart from income generation, the reasons why smallholders keep pigs vary in different regions. In Eastern Europe, they are kept mostly for home slaughter [2,8,9,10] and manufacture of traditional products that are consumed during the winter months and may sometimes be sold locally [6,11]. In some parts of the Pacific region, they are kept mainly for ceremonial or social purposes as well as an important source of income [12,13,14,15], and in many parts of Africa, as well as in Haiti, mainly as a passive asset, insurance policy, or “walking bank” to be sold when a need arises [16,17,18,19,20,21]. There is a need for better management of ASF in many types of smallholder farms and free-ranging systems. In the absence of widely available registered vaccines, prevention of ASF relies on the implementation of biosecurity measures. However, many of the recommended measures are not feasible in resource-limited settings. While ASF can have devastating effects on these pig keepers, the effects of measures either proposed to prevent or used to control outbreaks outweigh the damage caused by the disease itself.

Moreover, the spread of ASF virus (ASFV) in countries with important pork production has prompted calls for stricter legislation against smallholder pig production [1,22] or application of the same biosecurity requirements for all types of pig farms. This requirement has drastically reduced the number of smallholder farms; for example, in Estonia, from 696 in 2014 to 25 in 2017, in spite of the fact that they did not appear to be at higher risk of ASF than commercial farms [9]. In countries where smallholder pig farms are traditional and may represent a cherished hobby or be of local economic importance, there is a real danger that application of inappropriate policies and legislation could lead to loss of rare breeds and cultural identities [1], and considerable human suffering that is often overlooked [23,24].

In Africa, where the majority of pigs are kept in smallholder farming settings in areas where ASF is endemic, research to develop a better understanding of pig keeping and disease management in these specific settings has been undertaken over the last decades [25,26]. Challenges that have been identified include economic issues, such as the costs of housing and feeding pigs and access to veterinary services, but also sociocultural issues around limiting access by fencing properties, and customs associated with the disposal of dead pigs [27,28]. Much of the information gathered is applicable to smallholder farming wherever it occurs. In all smallholder settings, trade most often occurs through value chains with poor biosecurity that pose a high risk for the spread of ASFV [8,29,30,31]. Furthermore, information on the size, composition, and distribution of this sector is not always accurately known, since recordkeeping is often lacking.

While free-ranging of pigs occurs in smallholder settings in most developing countries, there are important geographic differences in the role of wild-pig family members in ASF. Transmission from the classic sylvatic cycle to domestic pigs that occurs in eastern and southern Africa is almost exclusively indirect, mainly via infected ticks of the *Ornithodoros moubata* complex picked up by domestic pigs sharing space with warthogs (*Phacochoerus africanus*). Limited direct transmission from acutely experimentally infected bushpigs (*Potamochoerus larvatus*) to domestic pigs has been demonstrated [32], as well as limited infection of domestic pigs fed liquidized offal that included lymph nodes from acutely experimentally infected young warthogs [33]. Under natural conditions, warthogs and bushpigs harbour little if any ASFV so are unlikely to shed virus [34,35,36], and the risk from their carcasses is accordingly very low. As in most regions, ASF in sub-Saharan Africa is mainly driven by circulation of ASFV in domestic pigs [37].

The involvement of Eurasian wild boars (*Sus scrofa*) in the transmission and spread of ASFV in the ongoing epidemic in Europe has resulted in a very different sylvatic cycle that has nothing in common with Africa and occurs between wild boars and the environment [38]. Wild boars are ancestral to and conspecific with domestic pigs, and share the same susceptibility to ASFV [39,40]. Unlike African wild suids, they are, therefore, efficient transmitters of the virus and also experience the same clinical signs, including high case fatality rates [40,41]. Recent analysis carried out by the European Food Safety Authority (EFSA) revealed that in 2019, there was a geographic overlap between cases in wild boar and outbreaks in domestic pigs in Lithuania, Poland, Latvia, Romania and Slovakia [42]. Prior to the introduction of ASF into the European Union (EU) in 2014, wild boars were not considered likely to sustain ASFV infection in the absence of domestic pigs. Recent studies have shown independent circulation in wild boars and the protective effect of confinement of smallholder pigs in areas with a high risk of wild boar contact [38,43]. However, a risk assessment for ASF in domestic pigs in the Samara district of Russia, where an equal number of cases of outbreaks in wild boars and domestic pigs were recorded, found that the movement of infected pigs and pork constituted the highest risk, with no indication of an association with wild boars [8]. Likewise, in the eastern parts of Europe (Romania, Bulgaria, eastern Poland) where smallholder pig farming is common and mainly involves poor people, ASF is spread within the domestic pig value chain, similar to other smallholder contexts around the globe [2,7]. Denstedt et al. [44] reported the occurrence of spill-over of ASFV from domestic pigs to wild boars in Laos, Vietnam, and Cambodia in smallholder settings, but not in the other direction. They also did not find evidence of the maintenance of ASFV within wild boar populations, while acknowledging the limited sensitivity of passive surveillance data. Given that first reports of an introduction of ASFV into the wider region (i.e., China) had only occurred in August 2018, it must also be taken into consideration that this study was based on data from 2019 and early 2020. The strongest evidence for sustained circulation in wild boars in Asia has emerged from South Korea, where outbreaks in both domestic pig farms and wild boars have occurred in areas bordering North Korea, mostly without apparent connection [45,46,47,48], and ASFV DNA has been reported from wild boar deathbeds [49].

The high risk of ASF to small populations of wild pigs in Asia, most of which belong to the genus *Sus*, has also been highlighted by conservationists [50], which makes it important to prevent their contact with potentially infected domestic pigs.

It is increasingly recognised that biosecurity measures need to be developed in partnership with the pig owners and other stakeholders in the value chain to ensure that they are financially feasible, culturally acceptable, and, thus, implemented [23,27,30,51,52,53,54,55,56,57]. The arrival of ASF in Eastern Europe, the Asia-Pacific region and, most recently, the Caribbean, has provided a new impetus for research that focuses on the specific circumstances and possibilities for pig keeping and disease management in the smallholder sector. The main focus of recent research on ASF has, however, been on vaccine development and on a better understanding of the antigenic determinants and molecular genetics of ASFV. This is understandable and important, considering the impact that ASF has on commercial pig production and international trade. It is nevertheless believed that the benefits of those advancements will vary in the smallholder sector, depending on limitations such as genotype variation versus homologous vaccine protection, thermostability, costs, and practical applications [21,53,58]. There is also a growing body of literature on antiviral substances that are active, at least in vitro, against ASFV, but this does not appear to offer short-term solutions for management of ASF in the resource-limited smallholder sector. These developments that are primarily relevant for medium- to mega-scale pig production have, therefore, not been included in this study.

This review provides an update on recent research that contributes to our understanding of the smallholder pig sector and the value chains that serve it, to inform more eco-socially acceptable approaches to managing the disease in that sector. It also provides science-based evidence to support specific interventions such as alternative approaches to assuring safety of affordable feed and carcass disposal, more rapid diagnosis of outbreaks and more measured, and sensitive approaches to outbreak response in resource-limited smallholder and backyard pig farms.

## 2. Approach to the Study

The update covers research published from 2018 to 2022, and was performed as a narrative review of articles covering the subjects of managing ASF in smallholder settings. Articles were identified through searches of bibliographic databases of scientific literature (Google Scholar, PubMed, Web of Science). Additional publications not identified from these databases, for example, those published before 2018 or in early 2023, were also included.

There are many definitions of smallholder pig farms, based variously on the number of pigs kept, the purpose for which the pigs are kept, and the production system. In this review, we focus on smallholder pig farming in resource-limited settings, where the greatest challenges for ASF management are to be found. In these settings, the herd size is usually relatively small, and pig production is often a secondary activity [14,16], although it is sometimes reported to be an important source of income [15,23,59].

While the largest numbers of resource-limited smallholder farms are found in LMICs in Africa, and the Asia-Pacific, central American, and southern American regions, the study recognises that ‘resource-limited’ is a relative term, and smallholder operations known as backyard pig farms are also common in some member states of the EU, particularly in less affluent countries in eastern Europe. These face many of the same challenges as smallholder pig farmers in LMICs, including an increased risk of ASF.

## 3. ASF-Related Challenges Facing Smallholder Pig Farmers

Poor people who keep a few pigs in smallholder settings face multiple and complex livelihood and farming challenges. Many animal health problems are often directly related to or aggravated by poverty and its direct consequences [55]. This has been described by Ebata et al. [55] as a vicious cycle where poor farmers cannot invest in their farming and in preventive measures, such as biosecurity, and, therefore, are more vulnerable to, and disproportionately affected by, animal diseases. Disease outbreaks and subclinical disease occurrence incur treatment costs, loss of animals, low production, and low income, with people falling even deeper into poverty as the final consequence. In this regard, animal diseases impact negatively on family livelihoods in a similar way to other disasters, and with more and more severe impact for poor and vulnerable people than for those who are better off [60]. In the case of ASF, the negative consequences of disease additionally include those of the control measures: culling of affected, in-contact, or at-risk pigs as well as restrictions of pig movements and trade. In LMICs, compensation for loss of animals or income caused by control measures is often inadequate or non-existent, exacerbating the losses from the disease and acting as disincentives for reporting suspicions of disease [61,62]. Given that disease reports initiate control measures and that the control measures applied are efficient for restricting the disease, failure to report suspected outbreaks will sustain disease spread and maintain the vicious cycle of poverty and disease [55].

Lack of awareness about ASF has frequently been mentioned as a real or perceived risk factor for ASF spread, particularly in smallholder settings with limitations in pig owners’ scientific knowledge of, for example, pathogens [27,28,63]. Recent research has, however, revealed that knowledge of ASF and of what is needed to prevent it will not necessarily result in improved implementation of biosecurity [64]. Factors that govern implementation have instead been identified as the need to prioritize the family livelihood over other expenses, biosecurity costs, feasibility of measures, access to pig feed, as well as access to and quality of veterinary services [27,64]. Veterinary services are generally scarce in low-income countries, especially in rural areas [65,66,67]. In addition, both private and governmental veterinarians prefer to serve better-off clients with larger farms and more possibilities to pay for the services, and poor smallholders are, therefore, particularly badly serviced [68,69]. Community animal health workers (CAHWs, a category of service providers whose education can vary from none to up to several years) can be a useful complement to licensed veterinarians in poor, rural settings with low coverage of veterinary services [70,71,72]. However, as the education and knowledge of CAHWs vary greatly, and as most will be named as “the vet” regardless of their official training, it is very difficult for smallholders to judge the importance of advice and treatments they receive and pay for [28]. Antibiotics or vitamins are, for example, frequently suggested for treating ASF, as there are several plausible differential diagnoses, adding costs without solving the problem [66] (Arvidsson et al., unpublished data). There have also been issues of sustainability without external financing [71].

## 4. Estimating the Impact of ASF on Smallholder Pig Farmers

Information on the impacts of ASF in the smallholder sector is scarce. There are several reasons: firstly, the economic losses might appear insignificant when compared to the losses on large commercial enterprises but still be severe at the household level, and secondly, the social and cultural impacts are difficult to quantify. Recently, some tools have been developed to gain a better understanding of the socioeconomic impact of ASF in smallholder settings. The OutCosT tool developed by the Food and Agriculture Organization of the United Nations (FAO) is a spreadsheet-based tool that can be applied across different pig-farming systems [73]. In Timor-Leste, a system dynamics approach was used to model the effects of the ASF outbreak on the pig value chain [74]. The approach includes developing a model in participation with stakeholders, including community members, via a process termed spatial group model building. It enabled the identification of potential leverage points where an intervention could mitigate the socioeconomic impacts of an ASF outbreak [74]. Finally, a tool developed by a multi-disciplinary team in Australia to evaluate the socioeconomic effects of livestock diseases on smallholders (Socioeconomic and Livelihood Impact Assessment—SELIA) was applied to ASF in the Philippines [23], revealing the complexity of the impacts of ASF and control measures. The SELIA framework integrates the concept of sustainable livelihoods and includes stakeholders throughout value chains which were analysed using a systems approach. Among the many socioeconomic impacts that emerged, two recommendations that stood out as particularly important were better communication and building trust amongst stakeholders, and a more humane ‘One Welfare’ approach to outbreak management [23].

## 5. Mitigating Well-Documented Risks

### 5.1. Focus on Biosecurity

Inadequate biosecurity is widely associated with ASF outbreaks on smallholder pig farms in developing countries [5,13,30,75,76,77,78]. As described by Oļševskis et al. [79], the epidemiological investigation of 28 ASF outbreaks in backyard pig farms in Latvia revealed that basic biosecurity measures, change of footwear, outer clothing, and disinfecting were not followed. The low level of biosecurity in backyard farms and the traditional particularities of pig keeping in Romania have facilitated the introduction of ASF in many backyard farms over a short period of time [42].

Biosecurity measures to protect pigs from ASF are intended to prevent direct and indirect contact between infected pigs or items and naïve pigs: confinement of pigs, ensuring that pigs introduced into the herd are healthy and free of ASF, restricting access of people to pigs, and ensuring the safety of feed [53,64,80,81,82]. Another paramount biosecurity measure is safe disposal of carcasses [76,83,84]. Implementing these measures requires an understanding of why they are needed, investment of money, effort, and time, all of which may be in short supply, especially in poor smallholder contexts [23,64,76,80]. Due to costs and limited access, providing pig feed is an overarching challenge hindering many poor smallholders from keeping pigs confined and, thus, for implementing almost all other biosecurity measures [27].

Even in countries where ASF is endemic, it remains a rare event for the individual smallholder, whereas, for example, a heavy parasite burden and other challenges, such as access to feed, are continuously present [85,86]. Recent research provides evidence that biosecurity implementation across the value chain can be improved by paying more attention to smallholders’ own disease priorities and engaging all stakeholders in the value chain to increase ownership of the disease and its control [54]. For this purpose, participatory approaches, consultation, and co-creation can be used to adapt control measures so that they are economically and practically feasible as well as socio-culturally acceptable [23,53,54,76,87,88].

### 5.2. Confinement

Confinement of pigs has been shown to be protective against ASF even in the midst of an active outbreak [4,13,27,89,90]. At the same time, confining pigs does not guarantee protection from ASF. For example, the cost of feeding confined pigs may result in the use of potentially unsafe swill, and the pigs remain at risk associated with the introduction of new pigs and contaminated fomites [4,80,91].

Montgomery [35] noted the protective power of confinement when he observed that ASF occurred in pigs that were free-roaming or not properly confined and, therefore, sharing the environment with warthogs and bushpigs, but permanently confined pigs were spared. During the next decades, confinement of pigs in double-fenced farms to ensure separation from wild suids was legislated in both Kenya and South Africa. However, in Angola, where ASF in domestic pigs also has a long history, outbreaks in settlers’ pigs were attributed to contact with free-roaming domestic pigs of local breeds, which appeared to be more resistant to ASFV [92]. When ASF arrived in West Africa, where the sylvatic cycle is absent, traditional free-ranging pig husbandry contributed importantly to the endemic establishment of the disease in most of the affected countries [89]. Confinement of pigs (preventing scavenging and mixing with pigs of different origins) is, therefore, recommended, regardless of the presence or absence of wildlife populations.

Outdoor, free-range farming of pigs is, furthermore, not exclusively associated with traditional and resource-limited smallholder pig farming. There are outdoor pig production systems that need diets based on acorns and chestnuts to produce the type of pork required for the finished, dry-cured product, for example, the Iberian pig outdoor production in Spain and Portugal and a similar system in Corsica, Bulgaria (East Balkan pigs), and Serbia (Mangalitza pigs) [93,94,95,96,97]. Organic and other outdoor pig farming is increasing due to consumer demand for raising animals humanely. A scientific opinion published by EFSA [94] on outdoor pig farming and ASF stressed that confinement of pigs should not be seen as non-negotiable due to the risk of ASF.

The constraint related to costs of construction materials may be overcome by identifying affordable local materials for constructing pens or fencing [13,80]. Breeding in traditional systems may also rely on free-roaming to enable encounters between boars and sows [3,86,98]. On the other hand, Arvidsson et al. [25] discussed how confining pigs helps reduce social tensions caused by free-roaming pigs potentially destroying crops, and how this argument for confining can be used regardless of whether biosecurity or disease control are considered priorities by smallholders. An ethnographic study exploring reasons for poor implementation of biosecurity among Ugandan smallholder farmers found that having pigs free-roaming was generally preferred [26]. This was partly due to the fact that smallholders perceived their pigs as an integrated part of the household, making it impossible to separate pigs from humans through confinement. In the study by Chenais et al. [54], smallholders reported how other community members regard them as either having become rich, or as trying to distance themselves from the rest of the community when confining pigs. This could also be a cause of social tension, especially in traditional societies where there is a strong communal identity and social control [25,26,99].

### 5.3. Restricting Entry and Protective Measures to Minimise Transmission via Fomites

Infected pigs are the most potent source of infection for susceptible pigs. When pigs are confined, purchase of additional pigs can introduce infection, and acquiring pigs from known safe sources and/or quarantining them for 15 days is recommended [21]. Quarantine is more often used in commercial farms and longer periods of 28–42 days are recommended [100,101,102]. This would be impractical for smallholder farmers, and in small premises with limited resources, effective quarantine is usually not an option [64]. Safe sources of new pigs might be difficult to find, with live markets and itinerant pig traders considered as posing a high risk [29,103]. It has been suggested that quarantine at community level may be possible in smallholder settings, for example, in a village if facilitated by the village head [13].

Sharing of boars is a common practice among smallholder farmers [3,4,10,76,88,104,105,106,107,108,109,110]. It is considered to be an unsafe practice because the status of the boars is not known, although it is unlikely that a clinically sick boar would mate. It has generally been recommended that artificial insemination would be a better alternative, because there was no scientific evidence that the virus could be effectively transmitted in semen, apart from a personal communication cited in a review that a researcher had successfully infected a sow via semen [111]. However, a recent study reported successful infection of gilts by insemination of semen from boars infected with the Estonia 2014 ASFV strain, which is considered to be moderately virulent [112]. There was a high abortion rate in the infected gilts, mostly when they developed a high fever, but replicating ASFV was detected in tissues of foetuses [112]. Transmission via semen from infected boar studs was considered to be a risk for commercial farms, but in practical terms, more evidence would be needed to determine the risk in the smallholder pig sector, where it is not widely used. Vertical transmission has also not been considered a high risk for ASFV, although it has been reported in the above study and in a retrospective report on vaccine development in Russia during the last century, after the introduction of ASFV into the Iberian Peninsula [113].

Finally, in some societies, pigs are received as gifts or as payment of debts or dispute settlement [12,98], or they may be received in exchange for goods and services [114]. In these circumstances, it may be difficult to refuse the pig or ensure the safety of the source, and the pigs should be quarantined.

Depending on the local setup, the same biosecurity aims (preventing direct and indirect contact between infected pigs or items and naïve pigs) can be achieved by restricting peoples’ access to pigs. If pigs are confined, restricting access to stables or sties adds one more layer to the total biosecurity protection. The guidelines established by the European Commission on ASF, among other biosecurity rules, recommend that no unauthorized persons/transport be allowed to enter the pig holding (stable), and that records be kept of people and vehicles accessing the area where the pigs are kept [83]. However, this may be difficult to implement in smallholder settings. Several studies have noted that restricting access to pigs can cause resentment, on the part of community members, at being excluded from premises and homesteads [27,64]. Introducing new biosecurity routines that interfere with how community members interact with each other’s pigs, such as measures intended to minimise transmission of fomites (washing hands, changing, or disinfecting footwear), might raise social tensions similar to those described for confinement above [54,64]. Community-based actions have been suggested as a way to overcome these kinds of social tensions around changed biosecurity behaviour [13,51,56,115,116]. A study to determine the effectiveness of community-engaged participatory interventions against cysticercosis caused by the pig tapeworm *Taenia solium* in resource-limited village settings, which is notoriously difficult, as it involves personal cultural practices, reported successful outcomes in two out of three villages [117].

### 5.4. Ensuring Safety of Feed

Pigs became domesticated because they were attracted to human settlements by the availability of edible waste [118]. Humans took advantage of the ability of pigs to convert waste food into high quality protein, and the result was a symbiotic relationship in which nothing was wasted. In view of pressure for more sustainable food production to reduce carbon emissions and contribute to the future survival of the planet [119], processing the vast quantities of food waste into safe and nutritious animal feed could make a highly significant contribution to sustainable pork production [120,121,122,123,124].

The ability of ASFV to persist for long periods in fresh or frozen pork, or in undercooked pork products, has been identified as posing a high risk to pigs that are allowed to scavenge or fed swill that might contain pork [30,53,79]. Secure confinement prevents scavenging, but often results in pigs being fed swill due to limited access to, and high costs of, commercial feed. Swill feeding is frequently mentioned as a probable cause of introduction of ASF into smallholder pig farms [17,19,30,108,125,126,127,128,129,130,131,132].

In the EU, there is a total ban on swill feeding [133], which was considered to be a practical and effective measure for ASF prevention by a panel of experts [126]. Unfortunately, that is only the case when all pig keepers have access to cost-effective alternatives for feeding their pigs. In lower income settings, swill feeding may be the only option, particularly in peri-urban settings where large amounts of catering waste may be freely available. In some settings, the backbone of smallholder pig keeping is that the pig effectively transforms household waste to pork. To accommodate this, and recognising the impossibility of enforcing prohibition of swill feeding in large numbers of smallholder pig farms, in several countries outside the EU, legislation permits the feeding of swill, provided that it has been processed in such a way as to destroy ASFV. However, the current recommendations for heat-treating swill are impractical in the backyard context, which is likely to lead to pig keepers continuing to feed unsafe swill; a risk assessment for ASFV in feed revealed that, even in the EU, illegal swill feeding occurred in three member states [133]. Article 15.1.22 of the World Organisation for Animal Health (WOAH) *Terrestrial Animal Health Code* requires that swill should be “maintained at a temperature of at least 90 °C for at least 60 min, with continuous stirring, or at a temperature of at least 121 °C for at least 10 min at an absolute pressure of 3 bar”. However, Article 15.1.23 indicates that inactivation of ASFV in meat requires “heat treatment for at least 30 min at a minimum temperature of 70 °C, which should be reached throughout the meat”, and this is the recommendation that is most often used for swill at farm level, including by the Food and Agriculture Organization of the United Nations (FAO) [134]. Recognising that backyard pig farmers are unlikely to have meat thermometers, this has often been replaced by boiling, as the only visual measure of approximate temperature, but the difference between 70 °C and ~100 °C is not taken into account, and the length of time remains the same or longer [21]. Smallholder pig farmers who feed swill because of affordability are unlikely to be able to afford cooking fuel for prolonged periods of heating and, equally, may not be able to spare 30–60 min to stir the mixture continuously. Montgomery [35] found that ASFV had lost all infectivity after being maintained at 60 °C for 10 min, demonstrated by inability to infect susceptible domestic pigs. A study by Plowright and Parker [135], using viral titre as an endpoint, indicated a marked decline in viral load in culture media after 10 min at 60 °C and no viable virus after 20 min at that temperature.

Until recently, specific information on the time required to boil swill has been lacking. Recent research in Thailand measured the reduction in ASFV titre at temperatures between 60 °C and 80 °C in three different swill formulations [136]. They developed a model based on decimal reduction time (D_T_) to predict complete inactivation of ASFV, measured as a 4–5 log titre reduction, at different temperatures. A marked reduction in titre occurred at 60 °C and the predicted time for complete inactivation at 90 °C was 4 min [136]. This indicates that a recommendation for boiling swill for 5 to 10 min, depending on volume, preferably with stirring, would be sufficient to destroy infectivity, given that the oral infective dose is generally accepted to be high. This practical recommendation would likely be accepted by many pig keepers, whom various studies have shown are eager to protect their pigs, provided the measure is feasible [52,64].

The use of commercial rations by smallholder and backyard farmers is less frequent than in commercial farms, but it is worth mentioning that available information about feeds contaminated with ASFV indicates considerable variability in the ability of different plant-based ingredients to maintain the virus, in the infective oral dose required, and in the length of time of storage at environmental temperature needed to inactivate the virus [133,137]. The use of dried pig blood as feed for pigs in China is a possible exception, as ASF genomic material has been reported, although virus could not be cultured [138]. However, liquid porcine plasma mixed with serum from an ASFV-infected pig and fed with a commercial feed to susceptible pigs for 14 days failed to result in infection [139]. Spray-dried porcine plasma particles contaminated with ASFV lost infectivity when stored for at least 14 days at room temperature [140]. The main concern for smallholder pig farmers using plant-based feed would be fresh forage fed to the pigs in areas where ASFV is circulating in wild boar populations [43,79]. A study using several field crops contaminated with ASFV-infected pig blood showed that storage at an environmental temperature of 20 °C (or above) for two hours resulted in reduction of the virus to a level at which it was not detectable [141]. Thorough washing of field collected forage, followed by drying, should also be helpful in removing sources of contamination (blood, saliva, faeces, urine).

### 5.5. Safe Disposal of Carcasses

Safe disposal of carcasses of pigs that die of ASF can be challenging in smallholder settings, as deep burial is arduous without heavy digging equipment [27]. There may, furthermore, be cultural challenges, such as prohibition of burial of dead animals because that is a rite reserved for humans, or because nothing that could be eaten may be wasted [21,27,64]. As a result, carcasses may be inappropriately disposed of in the open and along, or in, waterways, where they pollute the environment and can be accessed by scavenging pigs [13,23,75,142]. The ideal solution would be to turn the carcasses into something useful by recycling. In spite of the concept being abhorrent to people who can afford to choose what they eat and to eat well, and has accordingly been the subject of legislation in their culture, in some cultures, dead pigs are consumed as a matter of course, and provided they are thoroughly cooked and that uncooked remains are not accessible to pigs, this would appear to be a safe method of disposal [21]. However, trade in the carcasses of pigs that have died of ASF does pose risks, discussed further in the section on value chains.

Composting is increasingly used for mass disposal of carcasses during infectious disease outbreaks and has been proven to destroy several important animal viruses, including those that cause avian influenza, foot and mouth disease, porcine reproductive and respiratory syndrome, and porcine epidemic diarrhoea [143,144]. Most recently, Hoang et al. [83] demonstrated that composting destroys ASFV within three days, as demonstrated by cell culture, although viral DNA could be detected by PCR for the entire 90 days of the experiment. Composting is considered the most environmentally friendly carcass-disposal method [84]. In tropical and warm temperate climates, temperatures low enough to affect the composting system are unlikely. However, to offset the effects of cold temperatures, carcasses were ground up prior to composting and it was found that, despite unfavourable climatic conditions (rain, snow), adequate temperatures were reached and prevailed for long enough to inactivate the PED and PRRS viruses [144]. In smallholder settings, grinding may not be available, but breaking the carcasses up might be sufficient, as rapid degradation of whole carcasses, including bones, was reported, as well as higher temperatures reached in the compost heap as opposed to windrows [83].

### 5.6. Cleaning and Disinfection

Enveloped viruses, including ASFV, are among the least challenging pathogens for inactivation because they are susceptible to a wide range of disinfectants [145]. A number of recent studies have confirmed the efficacy of commonly used disinfectants against ASFV and provide details on concentration and contact times, as well as broadening the range to include citric and acetic acid, despite the wide pH tolerance range of ASFV [145,146,147]. Sodium hydroxide and calcium hydroxide remain disinfectants of choice in the presence of organic material [145]. Sodium hypochlorite (bleach) is highly effective against ASFV but not in the presence of organic material [145]. Removal of organic matter and cleaning is highly recommended before applying disinfectants, as this will destroy more than 90% of microorganisms [148]. However, if earth floors are involved, thorough cleaning is not possible but caustic soda or hydrated lime can be used [75]. These two chemicals have the added advantage of being widely available in LMICs where commercial disinfectants are unlikely to be affordable for smallholders. A recent experimental study was conducted to determine potential persistence of ASFV in soil from wild boar death beds after the carcass had been removed, using a range of different soil types spiked with ASFV-infected blood [149]. While ASFV genome could be detected throughout the four weeks of the experiment, stability of live infected virus varied according to soil type. Infectious ASFV was demonstrated in specimens originating from sterile sand for at least three weeks, from beach sand for up to two weeks, from yard soil for one week, and from swamp soil for three days. The virus was not recovered from two acidic forest soils. The application, for one hour at room temperature, of either 3.5% calcium hydroxide or 7.5% citric acid, resulted in complete inactivation of the virus in sand, potting soil, and blood [149]. The role of environmental contamination and the use of disinfectants around pig farms and slaughter facilities has been a matter of debate, and this study indicates that the rather widespread use of hydrated lime is effective in inactivating the virus in the environment.

Although cleaning and disinfection after an outbreak of ASF is recommended, a recent experimental study confirmed the findings of Montgomery [35] and Steyn [150] that infectivity in pig pens where pigs have died of ASF is short-lived even without cleaning and disinfection, as introduction of susceptible pigs resulted in infection after one day but not after three, five, or seven days [151].

### 5.7. Safe Value Chains—Can This Be Achieved?

Smallholder pig farming is often linked to value chains that pose a risk for the spread of ASF [31,106]. Network analysis has been used to trace the movements of pigs to where they are bought or sold in live markets or at the farm gate and evaluate the risks of spread of ASF [11,107,152,153,154,155,156,157]. Network analysis has also been used in commercial pig value chains to predict risk of ASF and plan risk mitigation strategies in China and USA [158,159].

Live markets where pigs are sold are common in many countries. The risk of ASF is increased when pigs from different sources are assembled [29,103,106,142,160]. The source of outbreaks outside the ASF-controlled area in South Africa in 2012 was a facility where live pigs were sold by auction [161] and infection was detected in a proportion of pigs sampled at four live markets in Nigeria [29]. Closure of the sales venues in South Africa resulted in illegal sale of pigs turned away by the closed facilities (state veterinarian, Gauteng, personal communication, February 2012). A later study showed that pigs from the controlled area were regularly moved illegally to an auction facility outside the area where prices were higher [152]. Documented outbreaks in Lusaka, Zambia, were traced to a market where pigs from rural areas were slaughtered [162]. The market was an informal one that was slaughtering pigs illegally [163]. Pigs may be moved over long distances to live markets [29,154] or they may be local [11,157], in which case, movement of pig products may pose a higher risk [157]. While live markets are generally considered to pose a high risk for spread of ASF, they can also serve as points for awareness creation and diagnostic testing and, thus, contribute to better control [11]. Diagnostic testing as well as clinical surveillance would be valuable because selling of pigs during outbreaks to limit losses is commonly reported [27,106,142,164]. This includes trade in sick and dead pigs [52,98,106], a coping mechanism to offset losses due to ASF. Selling sick pigs was found to be a component of smallholder farmers’ pig health management strategy in northern Uganda [165]. Sale of pigs that appear healthy as soon as mortalities are experienced in the owner’s herd or in the community is, likewise, widely reported to take place in order to avoid major losses or culling [161,166]. Pigs incubating the disease are likely to be included, which explains why abattoir sampling surveys often reveal infection in pigs that were apparently asymptomatic at slaughter [160,167,168,169,170,171]. A drop in prices paid for pigs during outbreaks has been reported [31].

Unsafe slaughter resulting in environmental contamination, not only with blood but with butchery waste that can be accessed by scavenging pigs, has been flagged as a major risk for the spread of ASFV [52,103], whether from home slaughter, markets, or slaughter slabs. Home slaughter is commonly practised in many low-income settings [6,11]. While the meat is usually for home consumption, some of it may be sold or given to friends and relatives [6].

Like pig farmers, other value chain actors responded to education and awareness creation and appreciated information needed to mitigate the risk of spread of ASFV through trade, but implementation fell short of expectations, mainly due to economic imperatives [106]. Including all stakeholders in the value chain in community-based actions for disease control has been reported as a key to success, facilitating implementation through strengthened cooperation and community ownership of diseases and disease control [54].

A system dynamics approach with group model building demonstrated that ASF risk mitigation along value chains, through implementation of biosecurity measures, needs to be combined with the development of business hubs along the value chain for ASF control to be effective and profits from pigs to increase [31]. This approach should be applied more widely to develop safer and more profitable value chains for smallholder farmers. Guidelines for setting up clean value chains that reduce the risk of ASF to an acceptable level are available for smallholder pig production in Asia and can be more widely applied [172].

## 6. Appropriate Outbreak Responses and Starting Again

### 6.1. Early Detection of Outbreaks

Preventing major ASF outbreaks and managing the disease with minimal disruption to livelihoods depends on early detection of infection [134]. In the current epidemic in Europe, early detection of outbreaks in wild boars in new areas has proved paramount for control and eventual eradication of the disease [83,172,173]. In endemic areas, the time component becomes less urgent, but early detection will still serve to limit the extent of outbreaks. Because of the characteristics of ASF, with high and early (before the onset of antibodies) case fatality, passive surveillance (reporting of the suspected cases) is the most important tool for early detection in the current ASF epidemic, both for domestic pigs and wild boar [173,174,175]. In LMICs, current surveillance systems are often underfunded, inefficient, and dysfunctional due to, e.g., deteriorating administrative services, budget reductions, and lack of veterinary personnel [176,177,178,179]. Participatory disease surveillance (PDS) has been suggested as a way to facilitate (early) reporting [180]. PDS has been successful in many different settings and for different diseases but is mostly run on project bases and rarely integrated into national surveillance strategies [181,182,183,184]. Syndromic disease surveillance is an approach that enables reporting of health events by a range of stakeholders including lay persons [185]. Lately, smartphone-based applications have been developed for different surveillance purposes, including resource-limited settings and for ASF [62,186,187,188]. Successful applications build on a combination of community engagement, letting smallholders’ disease priorities drive the development of applications and modern technologies [189].

### 6.2. Improved Sampling Methods for Early Detection

Performing post mortem examinations in smallholder farming settings can be challenging and can result in excessive environmental contamination. There has been a focus on simplifying sampling and using non-invasive or less invasive techniques to acquire suitable samples for diagnosis. For sampling dead pigs, blood swabs were found suitable for detection of both ASFV DNA and antibodies [190]. Superficial inguinal lymph node samples compared favourably with samples from spleen [191] and contained more ASFV than nasal, pharyngeal, or rectal swabs [192]. Oral ropes have proven useful for early collection of samples from live pigs, before clinical signs appear [193].

### 6.3. Diagnostic Tests for Field Application

The WOAH *Manual of Diagnostic Tests and Vaccines for Terrestrial Animals*, 2022, Chapter 3.9.1, describes the recognised international standards for ASF diagnosis. However, within countries there can be circumstances that make timely submission, processing, and testing of field samples difficult, which leads to delayed ASF response. Laboratories with adequate diagnostic capacity are not always available in LMICs, or may exist but not be functional. Although international donors have provided specialised laboratory equipment and vehicles to a large number of countries, the resources to employ adequate staff and maintain as well as operate equipment at immense expense may be lacking. Even when a functional laboratory is available, samples may need to travel long distances, which may be complicated by poor infrastructure, inadequate vehicle maintenance, shortages of fuel, inefficiency, or lack of couriers and by sample deterioration due to inability to maintain the cold chain, all of which lead to delays in obtaining results [194].

Due to the delays, attention has turned to pen-side, point-of-need, or point-of-care (POC) assays, which are inexpensive, field-deployable, and deliver highly reliable results. Questions have been raised about their efficacy, but in resource-limited smallholder farming settings, simple rapid tests may be appropriate in the context of on-farm epidemiological investigations and for wild boar carcass screening in the field. Field-deployed POC tests are rarely intended to replace traditional laboratory confirmation, and representative samples will still require laboratory confirmation and genetic characterisation for ASF outbreaks [194]. During ASF outbreaks, infected animals mostly die prior to the occurrence of antibodies, limiting sero-diagnostic use; therefore, POC assays are based on antigen (virus) detection.

Several screening tests or lateral flow assays (especially immunochromatographic assays) based on ASFV antigen detection, which are rapid, and do not require incubation, skilled technicians, or precision instruments, are commercially available (Table 1). These could thus be used by local veterinary services, where in many cases, first evidence of the disease is based only on clinical signs.

Table 1 is a summary of The OIE ASF Reference Laboratory Network’s overview of African swine fever diagnostic tests for field application [214] and other research publications. These LFD tests (Table 1) were indicated in experimental facilities to be most sensitive during the clinical phase of ASF, at the peak of viral replication [197,198]. This is adequate during acute outbreaks but will not always be the phase of infection in which field cases of ASF are reported and investigated.

All POCTs developed have not been applied in clinical samples in the field (due to biosafety regulations) but worked very well in the laboratory experiments. Outside the laboratory environment, in clinical samples from the field, suboptimal testing conditions can contribute to lower test accuracies than those reported by POC manufacturers, due to variations in temperature, humidity, operator ability, water and reagent quality, inadequate cold chain, and poor or non-existent quality assurance systems (Hobbs et al., 2021). However, a recent on-farm study confirmed the utility of LFD for diagnosis of ASF in clinically sick pigs [199]. Similarly, DNA extraction/releasing methods combined with different portable real-time PCR detection systems indicated that ASFV in field samples of clinically sick pigs can successfully be identified [207,208]. After POC testing in the field, a quick response by veterinary services will be required to ensure the community perception and acceptability of treatment or management decisions [194] for POC testing are successfully implemented.

### 6.4. To Cull or Not to Cull?

While stamping out all the pigs on ASF-affected and ‘dangerous contact’ premises, sometimes within a defined radius regardless of the health status of and real risk to the farms included, is considered key to eradicating ASF as quickly as possible, this approach is increasingly being questioned. Reasons for this include costs, ethical considerations, the resources required, the potential environmental impact of the disposal of very large numbers of carcasses, public disapproval, the trauma experienced by pig owners as well as people who have to carry out the slaughter, and the loss of valuable pig breeds, as happened in Haiti in 1978–1984 [18,23,24,215]. It has recently been reported, after a second incursion into Haiti in 2021, that the country has never fully recovered from the 1978–1984 ASF outbreaks and the drastic response [18]. There is also the ethical issue, when pre-emptive slaughter is included, of the sheer waste of high-quality protein from healthy pigs, when ordinary food wastage in high income countries is already a matter of considerable concern [120,216,217]. For example, during the classical swine fever outbreaks in 1997–1998 in the Netherlands, approximately 11.1 million pigs were destroyed, only 700,000 of which were infected, with more than 1 million healthy pigs destroyed pre-emptively and, due to a complete movement standstill, over 7 million healthy pigs killed for welfare reasons and destroyed because they could not be moved to an abattoir for slaughter [218]. The 1996 and 2014 ASF outbreaks in Côte d’Ivoire were eradicated by destroying all the pigs within the Greater Abidjan area in 1996 and a 10 km radius around the 2 foci in 2014, with compensation at 1/3 of market value [219,220]. In both cases, isolated and highly bio-secure farms within the designated radius were also depopulated, resulting in severe hardship for the farmers, including women who were beneficiaries of a project for poverty alleviation through modern pig-farming (MLP, personal observation, 2014). LMICs can usually not afford to pay adequate compensation for healthy pigs slaughtered and loss of income, which results in under-reporting, concealment, and illegal movement of pigs that can prolong the outbreak and increase the area of the outbreak [23,220]. Alternatives such as insurance schemes, soft loans, replacement of breeding stock in lieu of cash, and credits as a reward for investment in biosecurity have been proposed [23,74], as have alternatives to massive culling. The need for pre-emptive culling of pigs on all premises within a defined radius of the infected focus is highly questionable given that airborne transmission of ASFV has only been demonstrated over distances of a few metres within a shared confined space [221,222]. Alternative approaches include modified culling combined with quarantine and controlled marketing [89,134]. Modified culling that targets only infected herds or even infected animals is made possible by the observed slow spread of ASF, even in intensively farmed pig herds, as well as in a backyard farm, and has been confirmed in experimental studies [1,7,199,223,224,225,226,227]. After Ghana’s first incursion of ASF in 1999, only infected herds were culled, and healthy pigs were transported to a designated abattoir under veterinary permit and the meat processed and sold through a designated retail outlet [89]. The last ASF outbreak in Ghana was registered in February 2000 but, unfortunately, the disease was reintroduced from a neighbouring country in 2002 and, subsequently, became endemic [89]. Modified culling was used in South Africa for outbreaks that occurred outside the ASF controlled area in 2012 and 2020 [75,161]. Although this approach has been used quite widely in countries where generous financial compensation is not available, there is little published information on its effectiveness because countries wishing to provide a self-declaration of freedom to WOAH are reluctant to admit that they did not follow the conventional approach of massive culling. In the EU member states where compensation systems are in place, culling of all pigs in the ASF-affected pig farms is determined by the EU legislation; however, in certain situations, derogations can be applied [228]. A recent publication provides scientific evidence that modified culling protects livelihoods and achieves control of ASF, with, on average, more than 50 percent of the stock saved and only an 8-day extension of the time taken for implementation [225]. The approach is facilitated by the increasing availability of POCTs for on-farm testing [199].

Partitioning is an approach to minimising both spread of the virus and, therefore, culling on commercial pig farms that involves separation of the pig population into secure units or subpopulations to facilitate surveillance for early detection and minimise the risk of the whole herd becoming infected [229]. This approach was followed on a large commercial holding in China and, as a result, only half of a barn, representing 17.86 percent of the pigs in the facility, was lost [227]. While this approach targets commercial farms, the principle may be applicable in the smallholder sector as well. During widespread ASF outbreaks around Maputo, the capital of Mozambique, some backyard pig farmers managed to protect their pigs by strict confinement together with limiting access to the pigs to essential carers and feeding safe feed (MLP, personal observation). Similar protection of pigs, even in free-roaming systems, by farmers applying biosecurity measures and segregating their pigs, has been reported from an endemic region in Tanzania [103]. They were, in fact, applying the principle of partitioning. However, these farmers need to be protected by legislation that ensures that their pigs will not be culled pre-emptively during control efforts, as this is both unnecessary and inhumane. Massive culling was not practised in Mozambique due to inability to pay compensation, but policy must change in countries where the area culled is currently determined by a geographic radius and not by the status of farms within that radius or the availability of adequate compensation for healthy pigs culled [1].

Backyard holdings, in general, are, due to their small size and few animals, favourable for early detection of ASF, since sick and dead animals are usually spotted relatively early during an infection period. In large commercial farms, ASFV might circulate for several weeks before it causes a substantial increase in mortality and the disease is noticed. Timely identification of infected animals and holdings is the main prerequisite for applying a strategy where not all animals need to be culled (restrictive culling). The biology of the disease would accommodate this approach. Since the disease is not highly contagious, it does not spread rapidly [1,7,223,224]. However, due to the high virulence of ASFV, the lethality is very high, so that most infected animals become severely ill and die. This favours early detection, especially in the smallholder setup. The affected farms could be isolated immediately so that pigs on neighbouring smallholder farms do not have to be culled. The same principle could be applied within a farm if pigs are kept in different, well-separated pens. Nevertheless, to facilitate early detection and to avoid secondary spreading, it is crucial that after an outbreak, all neighbouring smallholder farms are inspected for the presence of ASF for several weeks until it is certain that the disease has not spread.

### 6.5. Starting Again

The question of whether it is permissible to retain pigs that survive an outbreak of ASF or whether they should be pre-emptively culled as probable long-term carriers of the virus is controversial. Successful eradication of ASF in Spain in the last century was attributed to a policy of killing all surviving pigs [230], although a study in Cataluña showed that all of the seropositive pigs culled in that province were virologically negative [231,232]. Recent experimental studies using the Netherlands ‘86 moderately virulent viral strain either failed to demonstrate transmission to in-contact pigs over a period of two months post recovery [233] or demonstrated a low rate of contact transmission over roughly the same period [234]. Although the latter study showed that ASFV transmission to 2 out of 12 in-contact pigs occurred under experimental conditions for approximately 8 weeks post-infection, it suggested that transmission from recovered pigs is not highly efficient. A longitudinal study on farms in Uganda that had experienced ASF outbreaks provided no evidence for persistence of ASFV in blood or serum of pigs [235], although no culling is undertaken in LMICs where ASF is endemic. The most significant study to date followed 14 gilts that recovered from infection with a highly virulent strain of ASFV in Vietnam [236]. The gilts were derived from a group of 70 convalescent weaner pigs that had shown mild or no clinical signs and survived an outbreak which had killed 409 of their cohorts in the same barn. The study period continued until the gilts were 497 days of age, during which time they and their litters were monitored for ASFV genome and antibodies. Briefly, all of the pigs were ASFV-positive on qPCR at the start of the experiment and for a period that varied from 42–70 days, after which all were negative in sera and nasal swabs and remained so throughout the experiment. The sows were bred using artificial insemination at 224 days of age. No vertical transmission of ASFV to piglets occurred. All the pigs remained positive for antibodies until the end of the experiment. Passive transfer of antibodies to piglets occurred, with levels remaining high at 21 days and declining gradually after weaning. The reproductive performance of the gilts was below average, with only 11 farrowing and one of the litters stillborn (PRRSV, CSFV, and ASFV were excluded as a cause), and litters were smaller. Nevertheless, the authors suggested that recovered sows could be suitable as replacement stock after an outbreak because they clear the virus after a relatively short period of time. Fears have been expressed that residual virus in tissues of surviving pigs might be reactivated by stress [237]. These fears seem groundless because there was no evidence that this occurred during 11 pregnancies and were, moreover, based on detection of ASFV protein in recovered pigs by immunohistochemistry after PCR failed to detect any ASFV DNA (Pornthummawat et al., 2021). The study by Oh et al. (2021) furthermore suggests that retaining pigs that recover from ASFV infection uneventfully is acceptable provided that they are not sold or slaughtered for anything other than home consumption for a period of at least two months after the outbreak.

### 6.6. Focus on the Main Drivers of ASF

An encouraging development has been the growing awareness of societal and cultural factors as major drivers of ASF. Domestic pigs are managed animals, and their managers are people. Ultimately, it is the pig owners who will determine whether their pigs will be able to have contact with wildlife, how they are housed, what they eat, the state of the environment in which they live, how and where they will be slaughtered or sold, and how to dispose of dead pigs, pig waste, or butchery material not to be consumed. In this regard, it becomes important to understand what influences animal husbandry and disease-control decisions, and how people can act depending on “soft factors” such as culture, tradition, knowledge, experience, and status in society, as well as “hard factors”, such as their financial situation and market, financial, feed, and veterinary access [28,53,55]. While many of these challenges are overwhelming, better management of ASF has been achieved through participatory approaches, community engagement, and public–private partnerships.

## 7. Conclusions

Despite ASF usually being described as a complex disease, research spanning a century has revealed certain characteristics that are helpful in managing ASF. These are that airborne spread is insignificant in the spread of ASFV between, and even within, premises where pigs are kept; that after a point of introduction, the virus spreads slowly; that the virus is inactivated in a relatively short time at 60 °C and above; and that, despite a wide range of temperature and pH tolerance, most disinfectants are capable of destroying ASFV. These features enable cost-effective prevention by relatively simple biosecurity measures, as well as removing the need to apply control measures that in the process also destroy livelihoods.

The global spread of ASF has reinforced our understanding of the ability of ASFV to circulate indefinitely in *Sus scrofa* populations (i.e., domestic pigs and Eurasian wild boars) without participation of alternative hosts and without convincing evidence for a long-term carrier state. Even in eastern and southern Africa where the ancient sylvatic cycle involving warthogs and ticks exists, major outbreaks in domestic pigs are linked to the domestic cycle. The smallholder pig sector, which is large in LMICs, has been disproportionately affected, and sometimes threatened with extinction. Until major social and structural problems that result in poverty are addressed, subsistence pig-farming will continue to be a source of much-needed income. To ensure that it fulfils this expectation, effective reduction of ASFV risk in both pig husbandry and pig value chains needs to be supported sustainably. This is best achieved through transdisciplinary community engagement to ensure participatory formulation of feasible, socially acceptable and effective interventions that are owned by the stakeholders and supported by both public and private sectors. Rather than focusing on ASF only, a holistic approach to management that results in more and healthier pigs, and that involves all the stakeholders in the pig value chain is more likely to gain traction. As women and children are primary carers for pigs, increased empowerment of these farmers with regard to animal treatment and management could protect them from shocks caused by diseases such as ASF and provide them with more resources to invest in their pigs.

It is understood that when proposing alternatives to traditional control measures for ASF in resource-limited settings, acceptance by the official veterinary services and the commercial pig sector is not guaranteed, particularly if their preferred goal is eradication. However, especially in countries where the disease is endemic, there is generally recognition that conventional approaches are not working, and practical, science-based approaches to risk mitigation provide an acceptable alternative.

When pigs are kept principally as a source of income, improved market access through reliable and safe value chains, and having a better product provides a strong incentive for better husbandry. Other incentives, such as having enough pigs to meet social and cultural obligations or simply maintain an ancient tradition, also contribute to improved husbandry to mitigate the risk of ASF. The recognition that human activities are largely responsible for the introduction and spread of ASFV suggests that progress towards better management of ASF is possible, even in the face of circumstances, such as poverty and climate change, that are largely beyond the control of the individual.

## Figures and Tables

**Table 1 pathogens-12-00355-t001:** Summary of point of care (POC) tests.

Test	Format	Sensitivity (Se) and Specificity (Sp)	References
Rapid screening test for ASFV based on p72 gene	Lateral flow	Se: Low to moderate (~68%), depending on sample quality Sp: High (98–100%)	[195,196,197,198,199]; https://ingenasa.eurofins-technologies.com/ (accessed on 10 January 2023).
Duplex pen-side detects antibodies and antigens specific to ASFV	Lateral flow	Se: Moderate antigen detection (66.7%) Sp: Antibody (97.5%) and antigen (98.1%)	[200]
Rapid screening test for ASFV based on p30	Lateral flow	Se: Moderate (~90%) Sp: High (100%)	[201,202,203]; https://Isybt.com/En; www.penchecktest.com/ (accessed on 10 January 2023).
ASFV Antigen Rapid test	Lateral flow	Se: Low to moderate Sp: Moderate	www.bionote.co.kr (accessed on 10 January 2023).
ASFV CRISPR/Cas12a-LFD	Clustered regularly interspaced short palindromic repeats (CRISPR) and CRISPR-associated (Cas) systems, named CRISPR/Cas12a, combined to develop a lateral flow	Se: Moderate to high Sp: High	[204]
Cas12a-based assay was combined with recombinase polymerase amplification (RPA) and a fluorophore-quencher (FQ)-labelled reporter assay for rapid and visible detection of the p72 gene of ASFV	DNA extraction, with the ability to be processed in only 30–40 min under visual inspection and fluorescence intensity	Se: High Sp: High	[205]
Cas12a/crRNA/ASFV DNA complex	ASFV DNA binding becomes activated and degrades a fluorescent single-stranded DNA (ssDNA) reporter	Se: High Sp: High	[206]
Portable real-time PCR assays	Real-time PCR	Se: High Sp: High	[207,208]
Recombinase polymerase amplification (RPA), or recombinase-aided amplification (RAA) combined with ASFV p72 gene with LFD	Real-time PCR combined with lateral flow dipstick	Se: High Sp: High	[209,210,211]
Loop-mediated isothermal amplification (LAMP) assay		Se: Moderate to high Sp: High	[212,213]

## Data Availability

Not applicable.

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
