# Peer review of "Innovative Research Offers New Hope for Managing African Swine Fever Better in Resource-Limited Smallholder Farming Settings: A Timely Update"

_pathogens, 2023, doi:10.3390/pathogens12020355_

Round 1

Reviewer 1 Report

Excellent paper with new prospectives taking into account the socio-economical constrains to facilitate the management African Swine fever in the smallholder farms which are among the ones that pose higher risks for disease transmission. It urges to have its recommendations applied in the field. I consider this paper as a deep review covering the most relevant epidemiological aspects with support from a large set (238 references cited) and updated scientific references. Although extensive It is well written and exciting to read. 

Author Response

We thank the reviewer for their time and effort in reviewing this manuscript and providing a positive review. 

Reviewer 2 Report

The manuscript is very well written, up-to-date, and provides practical, science-based guidance that can be of great help in managing the current epidemic, particularly in the context of non-intensive agriculture setting.

However, this approach must involve awareness and acceptance, at all levels, of having to live with the disease. In this regard, it is worth remembering that in the pig sector itself, interests are often conflicting. This does not mean that the approach and vision are wrong, but it would necessarily require, at least in some parts of the world, a change of strategy and rules, which must be acceptable to all or at least the majority.

Minor:

Line 783-787: The paragraph should be better explained.Consider revisiting the sentence to improve clarity.

Author Response

Thank your for the time and effort spent to provide a positive review, and the useful recommendations. They have been addressed as follows:

The following paragraph has been added to recognise possible acceptance problems by official veterinary services and the commercial pig sector:

It is understood that when proposing alternatives to traditional control measures for ASF in resource-limited settings, acceptance by the official veterinary services and the commercial pig sector is not guaranteed, particularly if their preferred goal is eradication. However, especially in countries where the disease is endemic there is generally recognition that conventional approaches are not working, and practical, science-based approaches to risk mitigation provide an acceptable alternative.

Lines 783-787 have been replaced as follows: 

This is best achieved through transdisciplinary community engagement to ensure participatory formulation of feasible, socially acceptable and effective interventions that are owned by the stakeholders and supported by both public and private sectors.

Reviewer 3 Report

This review describes the problems and solutions to ASF control in LMIC from a practical aspect. Among infected countries, the factors posing problems and their characteristics differ between developed and developing countries, thus it is obvious that disease control cannot be achieved through idealism alone. This review is a very detailed explanation of the points at issue in the field, supported by references. It shows the reader what the real problems are and provides useful information for other studies.

I have only one comment

6.3. To Cull or Not to Cull?

The question of culling all pigs in the vicinity of a reported infected farm has been raised by authors. One of the purposes of culling would be to remove those pigs that are infected but not yet showing clinical symptoms. Is information available on incubation periods for currently prevalent virus strains? I would recommend incorporating this perspective into this part if possible.

Author Response

Thank you for the time and effort taken to review this manuscript and your valid observation, which we have tried to address adequately.

Incubation periods are generally short but can vary from 3-10 days (this is more difficult to establish naturally infected pigs, as the time of infection is usually unknown), but the use of ropes to collect saliva samples while pigs are still well enough to be curious and active, i.e. not showing clinical signs, suggest that point-of-care tests will detect virus early enough to prevent spread by pigs that are shedding virus, and a sentence has been added about the increasing availability of such tests to support partitioning or limited culling. 

Reviewer 4 Report

African swine fever (ASF) is a highly devastating viral disease affecting domestic and wild pigs and is responsible for serious global food crises and economic losses. Rapid spread of the disease to long distance and introducing to newer countries strongly reminded that the virus can reach any region due to our globally connected world. In resource limited country, culling is not the absolute strategy to eradicate ASF and is not economically viable. Therefore, modified culling practice can be introduced. Partitioning is an approach to minimising both spread of the virus and therefore culling on commercial pig farms that involves separation of the pig population into secure units or sub-populations to facilitate surveillance for early detection and minimise the risk of the whole herd becoming infected. The review coved updated finding in support of Zoning and compartmentalizion as an alternative in resource limited country with majority raise pig in backyard farming system. It was specifically pinpoint the risk associated with spread of ASF outbreak and dicussed possible solution supported with finding. However, limited data on organized record keeping in ASF study. Hence, authentication required more proven data.

Author Response

Thank you for taking the time to review this manuscript and for a positive review. We have added a sentence to indicate that the smallholder pig sector is generally not well documented, which results in some uncertainties about approaches taken to control and their results.